# Protective Effect of Pioglitazone on Retinal Ganglion Cells in an Experimental Mouse Model of Ischemic Optic Neuropathy

**DOI:** 10.3390/ijms24010411

**Published:** 2022-12-27

**Authors:** Ming-Hui Sun, Kuan-Jen Chen, Chi-Chin Sun, Rong-Kung Tsai

**Affiliations:** 1Department of Ophthalmology, Linkou Chang Gung Memorial Hospital, Taoyuan 333423, Taiwan; 2College of Medicine, Chang Gung University, Taoyuan 333323, Taiwan; 3Department of Ophthalmology, Keelung Chang Gung Memorial Hospital, Keelung 20401, Taiwan; 4Institute of Medical Sciences, Tzu Chi University, Hualien 970374, Taiwan; 5Institute of Eye Research, Hualien Tzu Chi Hospital, Buddhist Tzu Chi Medical Foundation, Hualien 970473, Taiwan

**Keywords:** diabetes, ischemic optic neuropathy, pioglitazone

## Abstract

The aim was to assess the protective effect of pioglitazone (PGZ) on retinal ganglion cells (RGCs) after anterior ischemic optic neuropathy (AION) in diabetic and non-diabetic mice. Adult C57BL/6 mice with induced diabetes were divided into three groups: group 1: oral PGZ (20 mg/kg) in 0.1% dimethyl sulfoxide (DMSO) for 4 weeks; group 2: oral PGZ (10 mg/kg) in 0.1% DMSO for 4 weeks; and group 3: oral DMSO only for 4 weeks (control group). Two weeks after treatment, AION was induced through photochemical thrombosis. For non-diabetic mice, adult C57BL/6 mice were divided into four groups after AION was induced: group 1: oral DMSO for 4 weeks; group 2: oral PGZ (20 mg/kg) in 0.1% DMSO for 4 weeks; group 3: oral PGZ (20 mg/kg) in 0.1% DMSO + peritoneal injection of GW9662 (one kind of PPAR-γ inhibitor) (1 mg/kg) for 4 weeks; group 4: peritoneal injection of GW9662 (1 mg/kg) for 4 weeks; One week after the induction of AION in diabetic mice, apoptosis in RGCs was much lower in group 1 (8.0 ± 4.9 cells/field) than in group 2 (24.0 ± 11.5 cells/field) and 3 (25.0 ± 7.7 cells/field). Furthermore, microglial cell infiltration in the retina (group 1: 2.0 ± 2.6 cells/field; group 2: 15.6 ± 3.5 cells/field; and group 3: 14.8 ± 7.5 cells/field) and retinal thinning (group 1: 6.7 ± 5.7 μm; group 2: 12.8 ± 6.1 μm; and group 3: 15.8 ± 5.8 μm) were also lower in group 1 than in the other two groups. In non-diabetic mice, preserved Brn3A^+^ cells were significantly greater in group 2 (2382 ± 140 Brn3A+ cells/mm^2^, *n* = 7) than in group 1 (1920 ± 228 Brn3A+ cells/mm^2^; *p* = 0.03, *n* = 4), group 3 (1938 ± 213 Brn3A+ cells/mm^2^; *p* = 0.002, *n* = 4), and group 4 (2138 ± 126 Brn3A+ cells/mm^2^; *p* = 0.03, *n* = 4), respectively; PGZ confers protection to RGCs from damage caused by ischemic optic neuropathy in diabetic and non-diabetic mice.

## 1. Introduction

Diabetic retinopathy is the leading cause of adult blindness and is the most common complication of diabetes. It affects >90% of people with diabetes and ultimately leads to retinal edema, neovascularization, and vision loss in some patients [1]. Prolonged hyperglycemia in diabetes is associated with retinal microvasculopathy due to pericyte loss, acellular capillaries [2], capillary cell apoptosis, polyol-pathway-mediated basement membrane thickening [3], the release of vascular endothelial growth factor (VEGF) and proinflammatory cytokines (tumor necrosis factor (TNF)-α, interleukin (IL)-6, and IL-1β) [4], leukostatsis [5], and eventually, vascular hemodynamic changes, which are considered to play a pivotal role in diabetic retinopathy pathogenesis.

Non-arteritic anterior ischemic optic neuropathy (NAION) is the most common acute optic neuropathy in individuals > 50 years old [6,7]. NAION is multifactorial, and its systemic risk factors include arterial hypertension, diabetes, sleep apnea, ischemic heart disease, hyperlipidemia, and atherosclerosis [6]. Although diabetes is one of the most common risk factors for NAION [8,9], the role of diabetes mellitus (DM) in NAION severity is unclear [10,11,12]. Our previous study in a mice model of anterior ischemic optic neuropathy (AION) showed that diabetes increases the loss of retinal ganglion cells (RGCs), thus increasing the severity of retinal inflammation and damage, which can be reduced with short-term glycemic control [13]. In this study, we investigated the protective effect of long-term hyperglycemia control through treatment with pioglitazone (PGZ), a synthetic ligand of peroxisome proliferator-activated receptor-γ (PPAR-γ), on retinal tissues after AION inducement in mice fed a high-fat diet (HFD) with streptozotocin (STZ)-induced diabetes.

## 2. Results

### 2.1. PGZ Normalized Blood Glucose Levels and Did Not Change Body Weight

After rearing mice for 4 weeks on an HFD, their body weight increased (*p* = 0.479) (Figure 1A). Furthermore, 2 weeks after injecting these mice with STZ, their blood glucose increased gradually (Figure 1C), with a slight decrease in body weight (Figure 1B), and they became diabetic. Diabetic mice treated with 20 mg/kg PGZ had lower blood sugar levels (234.0 ± 10.0 mg/dL, *n* = 10) than diabetic mice treated with 10 mg/kg PGZ (271.2 ± 14.1 mg/dL, *n* = 10) and diabetic mice treated with 0 mg/kg PGZ (DMSO only) (368.0 ± 28.3 mg/dL, *n* = 10) (*p* = 0.0016; Figure 1D).

### 2.2. PGZ Preserved Retinal Thickness on OCT Measurement after AION in DM Mice

Since our preliminary study showed that the thinning of peripapillary retinal thickness at the posterior pole scan was more significant than the ganglion cell complex (GCC) thickness at AION W1 and AION W3 in diabetic mice without treatment of PGZ (Figure 2A), we evaluated the protective effect of PGZ on the preservation of peripapillary retinal thickness at AION W1. We found peripapillary retinal thinning was noted 1 week after AION induction in DM mice. Diabetic mice treated with 20 mg/kg PGZ showed lesser thinning compared with that in the other two groups (20 mg/kg PGZ group: 6.7 ± 5.7 µm thinning, N = 15; 10 mg/kg PGZ group: 12.8 ± 6.1 µm thinning, N = 15; and 0 mg/kg PGZ (DMSO only) group: 15.8 ± 5.8 µm thinning, N = 10; *p* = *0*.041; Figure 2B,C).

### 2.3. PGZ Reduced Apoptosis in RGCs 1 Week after AION in DM Mice

One week after ischemic optic neuropathy, RGC apoptosis was much lower in the retina of diabetic mice treated with 20 mg/kg PGZ (8.0 ± 4.9 cells/field, N = 8) than in that of diabetic mice treated with 10 mg/kg PGZ (24.0 ± 11.5 cells/field, N = 8) and diabetic mice treated with 0 mg/kg PGZ (DMSO only) (25.0 ± 7.7 cells/field, N = 6) (*p* = 0.013; Figure 3A,B).

### 2.4. PGZ Alleviated Iba1^+^-Activated Microglia Recruitment to Retina 1 Week after AION

Microglial cells were activated in the retina of diabetic mice 1 week after AION induction (Figure 4A). Activated microglia infiltration in the retina was lesser in diabetic mice treated with 20 mg/kg PGZ compared with that in the other two groups (20 mg/kg PGZ group: 2.0 ± 2.6 cells/field, *n* = 8; 10 mg/kg PGZ group: 15.6 ± 3.5 cells/field, *n* = 8; 0 mg/kg PGZ (DMSO only) group: 14.8 ± 7.5 cells/field, *n* = 6; *p* = 0.0016; Figure 4B).

### 2.5. PGZ Increased PPAR-γ Expression in Retina

To investigate whether the effects of PGZ are exerted through the PPAR-γ pathway, we performed immunofluorescence staining and found that PPAR-γ expression was barely detected in the retina of diabetic mice without AION and in the retina of AION eyes in diabetic mice without the treatment of PGZ (DMSO only). However, PPAR-γ upregulation after AION inducement was more significant in the retina of diabetic mice treated with 20 mg/kg PGZ than in that of diabetic mice treated with 10 mg/kg PGZ and the retina of normal eyes in diabetic mice treated with 20 mg/kg PGZ. Our results showed that PGZ treatment upregulated PPAR-*γ* expression in the retina of diabetic mice after AION induction (Figure 5).

### 2.6. PGZ Preserved RGCs after AION in Diabetic Mice

We counted Brn3A^+^ RGCs in the retinal whole mount of eyes harvested at 3 weeks after AION induction. The significant loss of Brn3A^+^ cells was observed after AION induction in diabetic mice treated with 0 mg/kg PGZ (DMSO only) and 20 mg/kg PGZ. However, the loss of Brn3A^+^ cells was significantly lesser in diabetic mice treated with 20 mg/kg PGZ than in those treated with 0 mg/kg PGZ (DMSO only) (DM-AION eyes treated with 20 mg/kg PGZ orally: 1788 ± 110 Brn3A^+^ cells/mm^2^, DM-control eyes treated with 20 mg/kg PGZ orally: 2441 ± 264 Brn3A^+^ cells/mm^2^, DM-AION eyes without PGZ treatment: 1491 ± 171 Brn3A+ cells/mm^2^, DM-control eyes without PGZ treatment: 2131 ± 298 Brn3A+ cells/mm^2^; *p* = 0.029; N =10 per group; Figure 6).

### 2.7. PGZ Preserved RGCs after AION in Non-Diabetic Mice

In order to elucidate whether the mechanism underlying the protective effect of PGZ on diabetic mice is antihyperglycemic-dependent or not, we fed C57BL/6 mice with a regular diet without STZ and counted the Brn3A^+^ RGCs in the retinal whole mount of non-diabetic eyes harvested at 4 weeks after AION induction. We found the preserved Brn3A^+^ cells were significantly greater in non-diabetic mice treated with 20 mg/kg PGZ orally every day (2382 *±* 140 Brn3A+ cells/mm^2^, N = 7) than in those without treatment (1920 *±* 228 Brn3A+ cells/mm^2^; *p* = 0.03, N = 4), in those treated with both 20 mg/kg PGZ orally every day and 1 mg/kg GW9662 (one kind of PPAR-*γ* inhibitor) intraperitoneally injected very other day (1938 *±* 213 Brn3A+ cells/mm^2^; *p* = 0.002, N = 4), and in those treated with only 1 mg/kg GW9662 intraperitoneally injected every other day (2138 *±* 126 Brn3A+ cells/mm^2^; *p* = 0.03, N = 4), respectively (Figure 7). Our results indicated that PGZ protects RGCs from AION insult through antihyperglycemic-independent effects and through PPAR-γ-dependent effects.

## 3. Discussion

Hyperglycemia could cause endothelial damage through the loss of pericytes and capillary apoptosis in a diabetic mouse model [1]. Moreover, observation under a scanning laser ophthalmoscope showed that in diabetic Ins2Akita mice [5], leukocyte velocity decreased, and the number of rolling leukocytes increased in the retinal arteriole, venule, and vein, and this phenomenon of leukostasis in the endothelial lining suggests inflammation in blood vessels under hyperglycemic status. Furthermore, hyperglycemia could promote vascular perfusion insufficiency from damage to the endothelium through the activation of the polyol pathway, the production of advanced glycation end products, an increase in oxidative stress, the upregulation of the protein kinase C-β pathway [8], and the overproduction of superoxide from the mitochondrial electron transport chain [14]. Producing reactive oxidative species through the overexpression of nuclear factor kappa B contributes to increased inflammatory cytokines, including TNF-α, IL-1β, IL-6, IL-8, vascular cell adhesion molecule-1, and intercellular adhesion molecule-1 [15], which in turn triggers leukocyte infiltration and causes vascular inflammation [16]. Thus, the optic nerve is vulnerable to ischemic insult in patients with diabetes. Some studies have suggested that diabetes increases NAION risk [8,9].

PPAR-γ, a member of the nuclear receptor superfamily, is a ligand-activated transcription factor that plays a crucial role in gene expression associated with various physiological processes including fat cell differentiation, glucose homeostasis, lipid metabolism, aging, and inflammatory and immune responses [17]. In the ocular tissues of mice, PPAR-γ is constitutively expressed in the neuroretina and retinal pigment epithelium [17]. PGZ, a type of thiazolidinedione, improves insulin sensitivity and lipid metabolism through the activation of PPAR-γ [18]; thus, PGZ has been used widely to normalize glucose levels in patients with type 2 diabetes. Moreover, PGZ regulates the lipid metabolism and reduces the levels of inflammatory mediators [19,20,21]. PGZ has shown protective effects for retinal ischemia/reperfusion injury [22], optic nerve crush injury [23], and normalized insulin signaling in diabetic rat retinas [24]. Our previous study showed that immediate blood sugar control with insulin treatment after AION induction in mice with STZ-induced diabetes reduced damage to the retinal structure [13]. In this study, we further investigated the effect of long-term hyperglycemia control through treatment with PGZ on retinal tissue protection after AION induction in mice with low-dose-STZ-induced diabetes fed an HFD, which mimicked type 2 diabetes through insulin resistance and partial damage to pancreatic β-cells [25]. Two weeks of PGZ (20 mg/kg) treatment could normalize blood glucose levels in HFD-fed mice with STZ-induced diabetes. Moreover, long-term PGZ treatment in HFD-fed mice with STZ-induced diabetes showed reduced apoptosis in RGCs, the decreased recruitment of Iba-1^+^-activated microglial cells, preserved retinal thickness on OCT measurement, and less loss of Brn3A^+^ RGC counts after AION, and these effects might have occurred through PPAR-γ activation in the retina, as demonstrated via immunofluorescence staining. Furthermore, PPAR-γ expression mainly on the RGC layer was higher in the retina of AION eyes in diabetic mice treated with 20 mg/kg PGZ than in that of diabetic mice without treatment or treated with 10 mg/kg PGZ. In order to elucidate whether the protective effect of PGZ on RGCs from AION insult is antihyperglycemic-dependent or not, we investigated the protective effect of PGZ on RGCs in non-diabetic mice. We found PGZ also preserved more RGCs from AION insult compared with the group without treatment, the group treated with both PGZ and the PPAR-γ inhibitor (GW9662), or the group treated with the PPAR-γ inhibitor alone. Our study suggested that PGZ could also protect RGCs from AION insult through antihyperglycemic-independent and PPAR-γ-dependent effects. To the best of our knowledge, this is the first study reporting the protective effect of PGZ on retinas after AION induction in diabetic mice and non-diabetic mice. Our findings provide insights into the potential therapeutic effect of PGZ on AION.

One limitation in this study is that we did not record the blood glucose levels of the non-diabetic mice treated with PGZ or the PPAR-γ inhibitor; it might be interesting to know if pioglitazone resulted in a decrease in blood glucose (or if the PPAR-γ inhibitor increased it) in non-diabetic mice in a future study. 

## 4. Materials and Methods

### 4.1. Animals

We used wild-type adult C57BL/6 mice weighing 25–30 g (Charles River, Hollister, CA, USA) housed in a temperature-controlled room with a 12 h light–dark cycle and with free access to food and water. All the animals were treated in accordance with the Statement of the Association for Research in Vision and Ophthalmology for Use of Animals in Ophthalmic and Vision Research. The mice were anesthetized with intraperitoneal injections of 50 to 100 mg/kg ketamine (Hospira Inc., Lake Forest, IL, USA), 2 to 5 mg/kg xylazine (Bedford Laboratories, Bedford, OH, USA), and 0.05 mg/kg buprenorphine (Bedford Laboratories). The pupils of the anesthetized mice were dilated with 1% tropicamide (Alcon Laboratories Inc., Fort Worth, TX, USA) and 2.5% phenylephrine hydrochloride (Akorn Inc., Lake Forest, IL, USA).

### 4.2. Diabetes Induction

Mice aged 4 to 5 weeks were fed a high-fat diet (HFD) (with 60% energy (Kcal/g) from fat, D12492, TestDiet, St. Louis, MO, USA) for 4 weeks and then intraperitoneally injected with low doses (40 mg/kg) of streptozotocin (STZ) (Sigma, St. Louis, MO, USA) in citrate buffer (pH 4.5) [26] for 3 consecutive days. Two weeks after diabetes induction, we obtained blood from the tail vein to test glucose levels using a basic blood glucose monitoring system (Accu-Check, Aviva Plus, Roche, Indianapolis, IN, USA), and mice with blood glucose levels consistently > 250 mg/dL were considered diabetic. The diabetic mice were divided into 3 groups: (1) mice fed orally with 20 mg/kg PGZ (Sigma) in 0.1% dimethyl sulfoxide (DMSO) with 10 mg/kg cellulose; group; (2) mice fed orally with 10 mg/kg PGZ (Sigma) in 0.1% DMSO with 10 mg/kg cellulose; (3) mice fed orally with DMSO with 10 mg/kg cellulose (control group).

### 4.3. Experimental AION

Two weeks after treatment, the blood glucose level returned to the normal limit in group 1. We induced AION in all the diabetic mice through photochemical thrombosis [27,28] following an injection of rose bengal (1.25 mM in phosphate-buffered saline; 5 μL/g body weight) into the tail vein by using a trans-pupillary laser in conjunction with a frequency-doubled Nd:YAG laser (Pascal, OptiMedica, Santa Clara, CA, USA) with 400 μm spot diameter, 50 mW power, 1 s duration, and 15 spots.

### 4.4. Spectral-Domain Optical Coherence Tomography

We performed spectral-domain optical coherence tomography (OCT) scans at baseline and 1 day, 1 week, and 3 weeks after AION induction by using Spectralis HRA + OCT (Heidelberg Engineering, GmbH, Heidelberg, Germany) [27,28,29]. We performed posterior pole scans (scan angle: 30° × 25°) to measure the peripapillary retinal thickness by using enhanced depth image (EDI) in the high-speed mode (a B scan consisted of 768 A scans, with an average of 9 frames/B scan) as well as 25 line scans (scan angle 25° × 15°) in the high-resolution mode (with an average 16 frames/B scan). The total retinal thickness, defined as the distance from the retinal nerve fiber layer to the Bruch’s membrane, was automatically segmented using Spectralis software, and the 1 or 3 mm diameter concentric circle grid with the optic disc in the center was measured. The OCT images of poor quality were excluded.

### 4.5. Immunohistochemistry and Morphometric Analyses

One week after AION induction, we performed intracardiac perfusion with 4% paraformaldehyde in phosphate-buffered saline, whole-mount retinal dissection, immunohistochemistry, and fluorescence microscopy (Nikon Eclipse TE300 microscope, Nikon Corporation, Tokyo, Japan) by using 4×, 10×, and 20× objectives (Nikon Corporation) and Metamorph software (Molecular Devices, Sunnyvale, CA, USA). To measure activated microglia, we performed morphometric analyses of fluorescence signals after the immunohistochemical analysis of paraffin-embedded retinal sections by using primary rabbit polyclonal anti-Iba1 antibody; these labels activated microglia (1:200 dilution; Wako Chemicals, Richmond, VA, USA). To analyze PPAR-γ expression, we used primary mouse monoclonal anti-PPAR-γ (1:200 dilution; Santa Cruz Biotechnology, Santa Cruz, CA, USA). Immunoreactivity was detected using fluorescein isothiocyanate (FITC)-labeled secondary antibody (Abcam, Cambridge, UK), and cell nuclei were counterstained with 4′-6-diamidino-2-phenylindole (DAPI). To count Brn3A^+^ RGCs, we performed whole-mount retinal dissection 3 weeks after AION and then stained them with primary mouse monoclonal anti-Brn3A antibody (1:200 dilution; Santa Cruz Biotechnology) and secondary goat antimouse IgG Alexa Fluor 568-labeled antibody (1:200 dilution; Life Technologies). All the retinal whole-mount preparations were mounted with 4′-6-diamidino-2-phenylindole (DAPI)-containing media (Vectashield, Vector Laboratories, Burlingame, CA, USA). To quantify Brn3A^+^ signals, we obtained 8 images (2 images with 4 quadrants, with each quadrant of 0.14 mm^2^) at 200× magnification and used custom-written ImageJ scripts to quantify and calculate the number of Brn3A^+^ cells/mm^2^.

### 4.6. In Situ TdT-Mediated dUTP Nick-End Labeling

The eyeballs of diabetic mice were harvested 1 week after AION induction and were sectioned along the vertical meridian to include the optic nerve head. For each mouse, two 3 μm thick retinal sections which included the ora serrata and optic nerve were stained by using a TdT-mediated dUTP nick-end labeling (TUNEL)-based kit (TdT FragEL; Oncogene, Darmstadt, Germany). The number of TUNEL-positive cells in each retinal section was obtained through the selection of 6 superior and inferior retinal areas, each 0.425 mm in length. First, we chose 2 segments 0.425 mm superior and inferior to the optic nerve head; second, we chose 2 segments 0.425 mm from the first 2 segments; and third, we chose 2 segments 0.425 mm from the second 2 segments. The total number of TUNEL-positive cells in these 12 retinal areas was averaged as a representative of the number of TUNEL-positive cells per eye sample.

### 4.7. Statistical Analysis

All data are presented as mean ± SD. We performed statistical analysis by using SPSS 23.0 (SPSS, Inc., Chicago, IL, USA), and statistical significance was defined as *p* < 0.05. We used the Wilcoxon signed-rank test for paired data and the Mann–Whitney *U* test for unpaired data. 

## Figures and Tables

**Figure 1 ijms-24-00411-f001:**
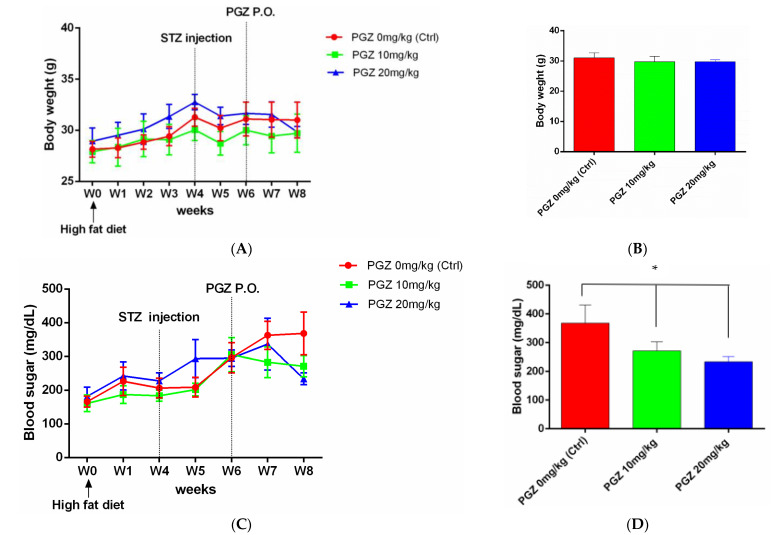
Blood glucose levels and body weights of mice. (**A**) Increase in body weight after high fat diet (HFD) rearing. (**B**) No significant difference in body weight between diabetic mice treated with 20, 10, and 0 mg/kg pioglitazone (PGZ). (**C**) Increase in blood glucose levels after streptozotocin (STZ) injection. (**D**) Diabetic mice treated with 20 mg/kg PGZ for 2 weeks had a lower level of blood sugar than those treated with 10 and 0 mg/kg PGZ (* *p* < 0.05).

**Figure 2 ijms-24-00411-f002:**
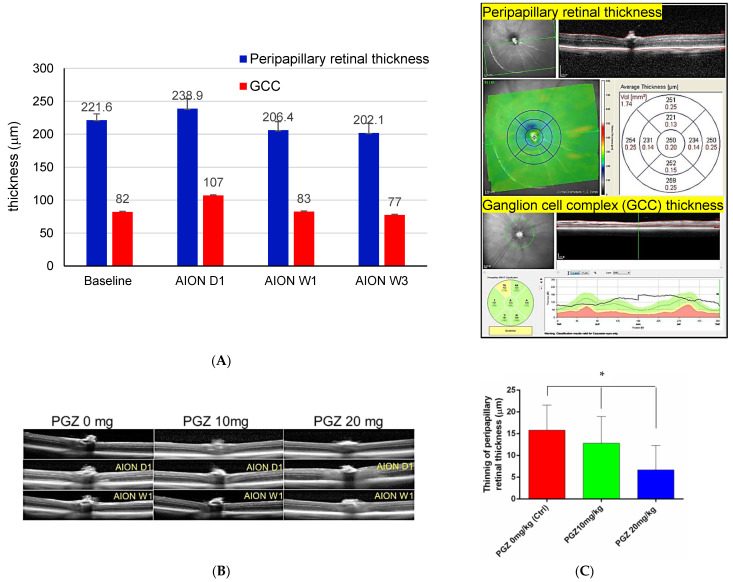
OCT measurement after AION in HFD feeding followed by STZ-induced diabetic mice. (**A**) Peripapillary retinal thickness using posterior pole scan and ganglion cell complex (GCC) thickness were measured through OCT on day 1 (AION D1), week 1 (AION W1), and week 3 (AION W3) in diabetic mice without treatment of PGZ (*n* = 17). The thinning of peripapillary retinal thickness was more significant than (GCC) thickness at AION W1 and W3. The up-right figure showed how the OCT machine measured the peripapillary retinal thickness and GCC thickness. (**B,C**) Less retinal thinning of peripapillary retinal thickness at posterior pole scan on AION W1 was noted in DM mice treated with 20 mg/kg PGZ than in the other 2 groups (* *p* < 0.05).

**Figure 3 ijms-24-00411-f003:**
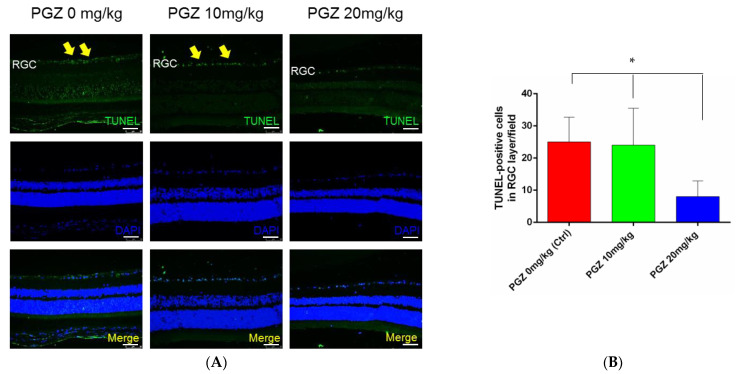
Apoptosis in RGCs 1 week after AION in DM mice. (**A**) Apoptosis in RGCs was evaluated through TUNEL staining. After ischemic optic neuropathy, apoptotic cells (yellow arrows) in RGCs were fewer in diabetic mice treated with 20 mg/kg PGZ than in those treated with 10 and 0 mg/kg PGZ (DMSO only). (**B**) Bar graph shows less apoptotic cells in RGCs in the retina of DM mice treated with 20 mg/kg PGZ compared with that in the other 2 groups (* *p* < 0.05) (scale bar: 75 μm; magnification: 200×).

**Figure 4 ijms-24-00411-f004:**
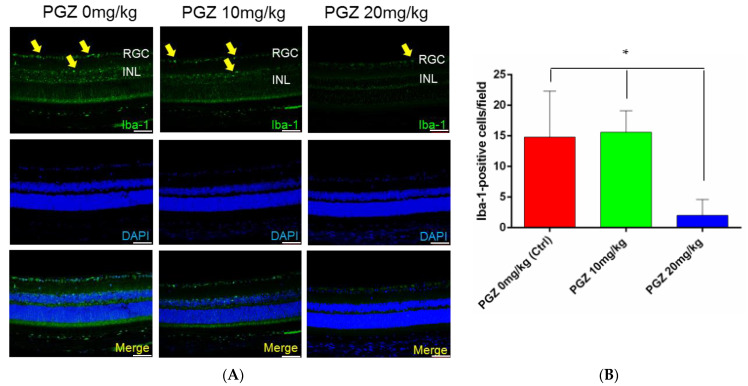
Recruitment of Iba1^+^-activated microglia to retina 1 week after AION inducement. (**A**) Microglial cells (yellow arrows) were activated in the retina of DM mice 1 week after AION inducement. (**B**) Bar graph shows less microglial cell infiltration in the retina of DM mice treated with 20 mg/kg PGZ compared with that in the other 2 groups (* *p* < 0.05) (scale bar: 75 μm; magnification: 200×).

**Figure 5 ijms-24-00411-f005:**
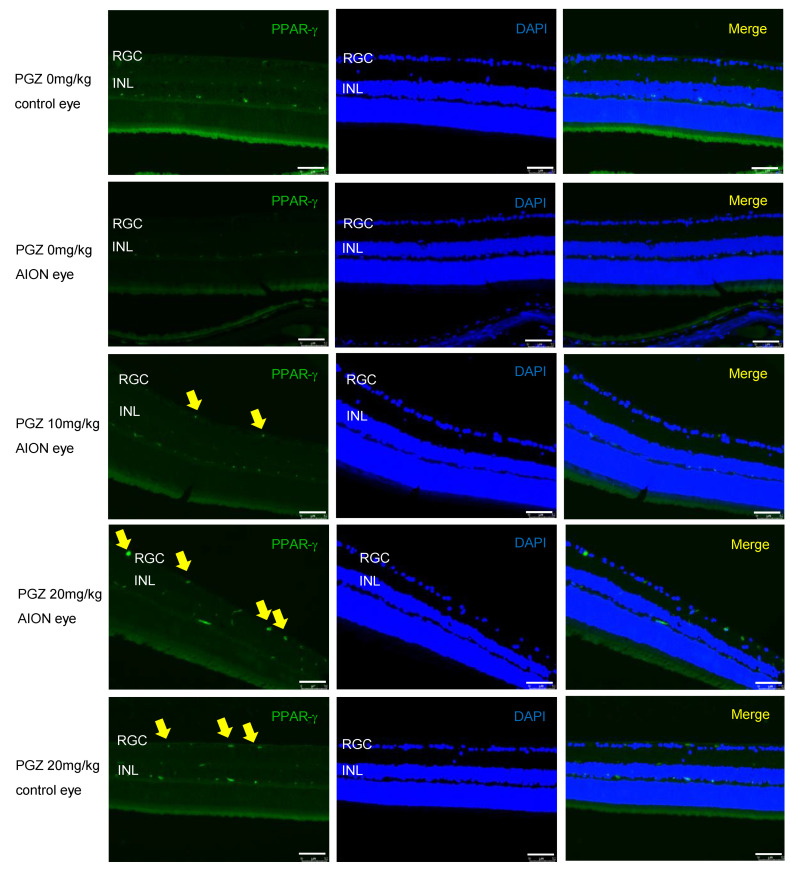
PPAR-γ expression in the retina. The representative figures show that PPAR-γ was barely detected in the retina of diabetic mice without AION and in retina of AION eyes in diabetic mice without treatment of PGZ. PPAR-γ (yellow arrows) expression was more significant in the retina of AION eyes in diabetic mice treated with 20 mg/kg PGZ than in that of diabetic mice treated with 10 mg/kg PGZ and retina of normal eyes in diabetic mice treated with 20 mg/kg PGZ (scale bar: 75 μm; magnification: 200×).

**Figure 6 ijms-24-00411-f006:**
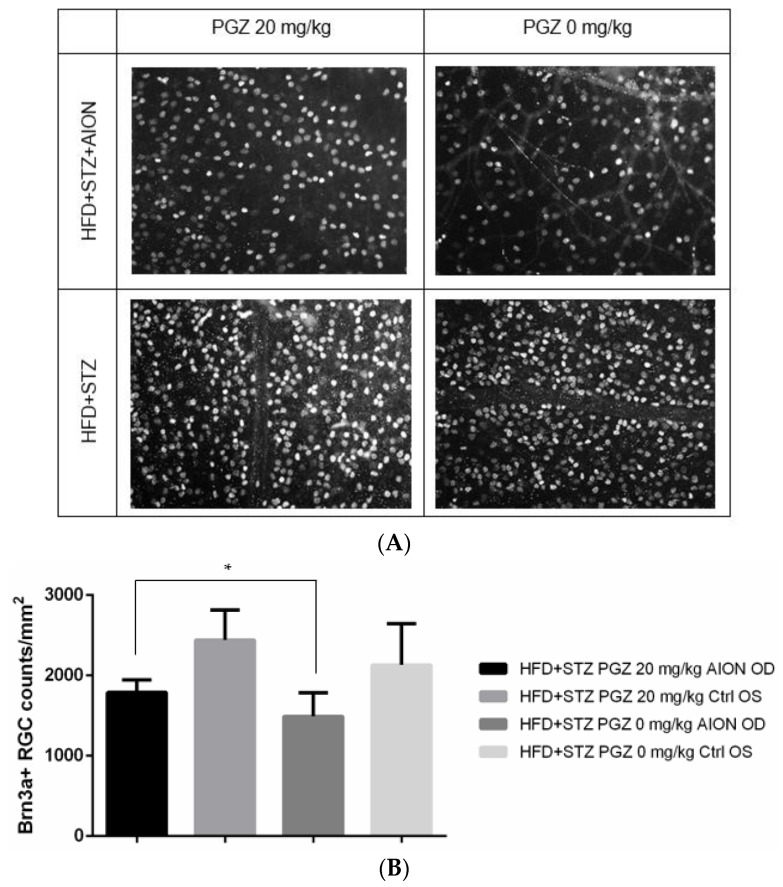
Brn3A^+^ RGCs in diabetic mice. (**A**) Representative images of immunostaining by using anti-Brn3A antibody in retinal whole-mount preparations of control and AION eyes of diabetic mice treated with 20 and 0 mg/kg PGZ 3 weeks after AION (magnification: 200×). (**B**) Bar graph showing average Brn3A^+^ cell counts in all 4 conditions 3 weeks after AION (* *p* < 0.05).

**Figure 7 ijms-24-00411-f007:**
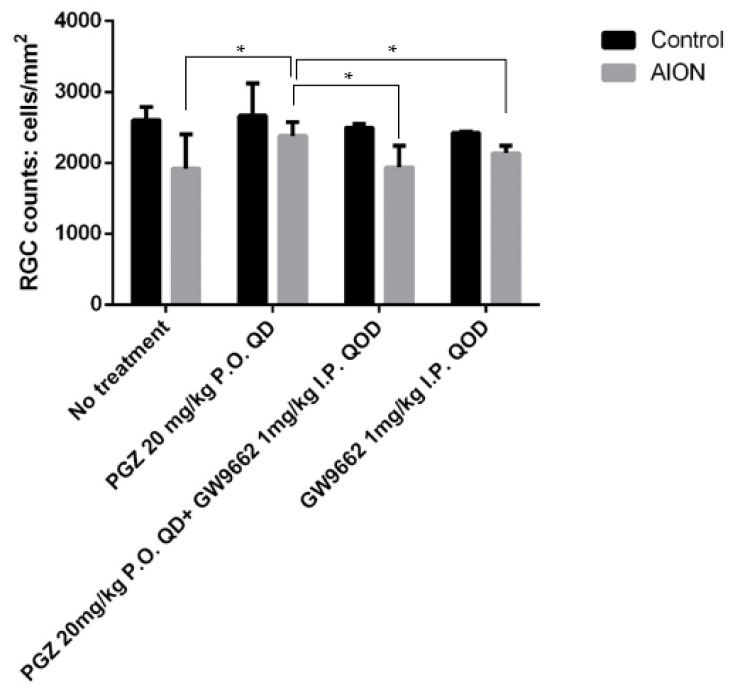
Brn3A^+^ RGCs in non-diabetic mice. Bar graph showed average Brn3A^+^ cell counts 4 weeks after AION induction in non-diabetic mice without treatment (oral DMSO only), non-diabetic mice treated with oral (PO) PGZ 20 mg/kg in 0.1% DMSO every day for 4 weeks, oral PGZ 20 mg/kg in 0.1% DMSO every day+ intraperitoneal (IP) injection of GW9662 (PPAR-γ inhibitor) 1 mg/kg every other day for 4 weeks, and intraperitoneal injection of GW9662 (PPAR-γ inhibitor) 1 mg/kg every other day for 4 weeks, respectively (* *p* < 0.05).

## Data Availability

Not applicable.

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
