# Peer review of "Protective Effect of Pioglitazone on Retinal Ganglion Cells in an Experimental Mouse Model of Ischemic Optic Neuropathy"

_ijms, 2022, doi:10.3390/ijms24010411_

Round 1

Reviewer 1 Report (Previous Reviewer 1)

Accepted for publication

Author Response

Thanks for reviewer 1's comment. Our manuscript was ever edited by professional English editing, we uploaded this certificate.

Reviewer 2 Report (Previous Reviewer 2)

I appreciate the authors making the effort to perform the additional experiment testing pioglitazone on non-diabetic mice. The inclusion of a PPAR-gamma inhibitor was a nice idea as well, and I think that this experiment provided some very intriguing results that will be interesting to follow up in the future.

A few minor things to address:

1. I would include this new result in the abstract, as one could argue it is the most interesting data in the study. You might even consider changing the title to “Protective Effect of Pioglitazone on Retinal Ganglion Cells in an Experimental Mouse Model of Ischemic Optic Neuropathy” since it appears that the effect is not limited to diabetes.

2. Section 2.6: did you happen to record the blood glucose levels of the mice in the different groups? Even though the mice were not diabetic, it might be interesting to know if pioglitazone resulted in a decrease in blood glucose (or if the PPAR-gamma inhibitor increased it).

3. Section 2.6: Can you describe the nature of these non-diabetic mice? Were they C57BL/6 mice fed a regular diet, or were they fed the high-fat diet but not induced with streptozotocin?

4. Figure 7: I see three asterisks (*), but only two brackets.  I’m assuming the third comparison to be indicated is PGZ to PGZ+ GW9662, but this needs a bracket to make this clear.

5. Discussion, line 372-373: I would shorten the final sentence to “Our findings provide insights into the potential therapeutic effect of PGZ on AION” and take out “in diabetes.” It seems there could be therapeutic efficacy regardless of diabetic status.

6. Discussion: last paragraph.  I would take out the final paragraph entirely. I don’t think that the first described limitation is that much of a limitation.  In my mind, RGC count is a much more reliable assessment of the neuroprotective effect than OCT thickness values. The second limitation, that “we didn’t test whether PGZ also has a protective effect on RGCs after AION in non-diabetic mice” is no longer a limitation, since you have now done that experiment.

Author Response

I appreciate the authors making the effort to perform the additional experiment testing pioglitazone on non-diabetic mice. The inclusion of a PPAR-gamma inhibitor was a nice idea as well, and I think that this experiment provided some very intriguing results that will be interesting to follow up in the future.

Answer: Thanks for reviewer’s comment.

A few minor things to address:

  1. I would include this new result in the abstract, as one could argue it is the most interesting data in the study. You might even consider changing the title to “Protective Effect of Pioglitazone on Retinal Ganglion Cells in an Experimental Mouse Model of Ischemic Optic Neuropathy” since it appears that the effect is not limited to diabetes.

Answer: Thanks for reviewer’s precious suggestion, we have changed our title into “Protective Effect of Pioglitazone on Retinal Ganglion Cells in an Experimental Mouse Model of Ischemic Optic Neuropathy”, which is marked yellow in the title page. Also, we revised the purpose in the abstract (line 15) and added the new results in the abstract as followed:

“For non-diabetic mice, adult C57BL/6 mice were divided into 4 groups after AION: group 1: oral DMSO for 4 weeks; group 2: oral PGZ (20 mg/kg) in 0.1% DMSO for 4 weeks; group 3: oral PGZ (20 mg/kg) in 0.1% DMSO + peritoneal injection of GW9662 (one kind of PPAR-g inhibitor) (1mg/kg) for 4 weeks; group 4: peritoneal injection of GW9662 (1mg/kg) for 4 weeks;” (line 19-23)

“In non-diabetic mice, preserved Brn3A+ cells were significantly greater in group 2 (2382±140 Brn3A+ cells/mm2, N=7) than in group 1(1920±228 Brn3A+ cells/mm2; P =.03, N=4), group 3 (1938±213 Brn3A+ cells/mm2; P =.002, N=4), and group 4 (2138±126 Brn3A+ cells/mm2; P =.03, N=4), respectively; (4) Conclusions: PGZ confers protection to RGCs from damage caused by ischemic optic neuropathy in diabetic and non-diabetic mice.” (line 28-32)

  1. Section 2.6: did you happen to record the blood glucose levels of the mice in the different groups? Even though the mice were not diabetic, it might be interesting to know if pioglitazone resulted in a decrease in blood glucose (or if the PPAR-gamma inhibitor increased it).

Answer: We apologize that we didn’t record the blood glucose levels of the non-diabetic mice. Since we want to investigate whether protective effect of PGZ on RGCs is also antihyperglycemic-independent, we treated C57BL/6 mice fed with regular diet rather than high-fat diet, which could induce hyperglycemia without the help of streptozotocin.  Therefore, we focused on the rescue effect of PGZ on RGCs after AION in non-diabetic mice to exclude the antihyperglycemic effect and didn’t measure blood glucose levels. But reviewer's point is very important, we add this point in the limitation of this study as followed:

“One of the limitation in this study is that we didn’t record the blood glucose levels of the non-diabetic mice treated with PGZ or PPAR-g inhibitor, it might be interesting to know if pioglitazone resulted in a decrease in blood glucose (or if the PPAR-g inhibitor increased it) in non-diabetic mice in future study.” (line 382-385)

  1. Section 2.6: Can you describe the nature of these non-diabetic mice? Were they C57BL/6 mice fed a regular diet, or were they fed the high-fat diet but not induced with streptozotocin?

Answer: Thanks for reviewer’s great point. We have described our mice were fed with regular diet without STZ as followed:

“we fed C57BL/6 mice with regular diet without STZ, and counted Brn3A+ RGCs in the retinal whole mount of non-diabetic eyes harvested at 4 weeks after AION.” (line 116-117)

  1. Figure 7: I see three asterisks (*), but only two brackets.  I’m assuming the third comparison to be indicated is PGZ to PGZ+ GW9662, but this needs a bracket to make this clear.

Answer: We apologize the bracket to indicate the difference between PGZ and PGZ+ GW9662 disappeared during image processing, we have put it back in the new Figure 7. 

  1. Discussion, line 372-373: I would shorten the final sentence to “Our findings provide insights into the potential therapeutic effect of PGZ on AION” and take out “in diabetes.” It seems there could be therapeutic efficacy regardless of diabetic status.

Answer: Thanks for reviewer’s valuable opinions, we have revised this sentence, which is marked in yellow (line 380-381).

  1. Discussion: last paragraph.  I would take out the final paragraph entirely. I don’t think that the first described limitation is that much of a limitation.  In my mind, RGC count is a much more reliable assessment of the neuroprotective effect than OCT thickness values. The second limitation, that “we didn’t test whether PGZ also has a protective effect on RGCs after AION in non-diabetic mice” is no longer a limitation, since you have now done that experiment.

Answer: We thank reviewer’s delicate review and give us a very useful suggestion. We have taken this part out, but no measurement of blood sugar level of non-diabetic mice treated with PGZ or PPAR-g inhibitor as mentioned in Question#2 is our limitation, we describe this limitation in last paragraph (line 382-385).

This manuscript is a resubmission of an earlier submission. The following is a list of the peer review reports and author responses from that submission.

Round 1

Reviewer 1 Report

Ming-Hui Sun et al. presented a study on the Protective Effect of Pioglitazone on Retinal Ganglion Cells in an Experimental Mouse Model of Ischemic Optic Neuropathy With Diabetes. The manuscript is well written and obtained results are interesting. However, several points need to be clarified in order to improve the readability of the manuscript.

  1. Pioglitazone is well-absorbed, with a mean absolute bioavailability of 83%and reaching maximum concentrations in around 1.5 hours if given orally as such but the authors used toxic solvent DMSO for pioglitazone administration orally, why not directly suspended into dispersing agent? Also, DMSO causes cytotoxicity as shown in cell line study results, which may not give true results of cytotoxicity of pioglitazone.
  2. As shown in Figure 1 A and 1C clearly, why the difference in body weight and blood sugar level in the post-treatment week, while it should be same almost during the start of the study?
  3. what was the body weight range (g) of animals that used in study, mention in study protocol? 

Author Response

Reviewer 1#

  1. Pioglitazone is well-absorbed, with a mean absolute bioavailability of 83%and reaching maximum concentrations in around 1.5 hours if given orally as such but the authors used toxic solvent DMSO for pioglitazone administration orally, why not directly suspended into dispersing agent? Also, DMSO causes cytotoxicity as shown in cell line study results, which may not give true results of cytotoxicity of pioglitazone.

Answer: Thanks for reviewer 1’s concerns about why the pioglitazone (PGZ) was dissolved in dimethyl formamide (DMSO) in our study.  According to the data sheet from Sigma-Aldrich website, PGZ was soluble in DMSO; and practically insoluble in water [O'Neil, M.J. (ed.). The Merck Index - An Encyclopedia of Chemicals, Drugs, and Biologicals. Cambridge, UK: Royal Society of Chemistry, 2013., p. 1383].  Also, several prior studies in vivo and vitro dissolved PGZ in 0.1-0.5% DMSO (Zhu et al. 2013, PLos one; Hu et al. 2012, J Surg Res; Thakran et al. 2014, Invest Ophthalmol Vis Sci).  Our protocol also referred to the previous study dissolving PGZ in 0.1% DMSO and 10mg/kg cellulose for oral gavage feeding (Majithiya et al. 2005, Cardiavascular Research).

  1. As shown in Figure 1 A and 1C clearly, why the difference in body weight and blood sugar level in the post-treatment week, while it should be same almost during the start of the study?

Answer: Thanks for reviewer 1’s comment. As shown in Figure 1A and 1C, We fed animals with high fat diet for 4 weeks (W0-W4) before STZ injection. During these periods, we weighed animals very week but didn’t check blood sugar every week because blood sugar level during these periods was not the main issue and we hoped to preserve more healthy tail vein. However, we did measure blood sugar very week after injection of STZ and even after oral gavage feeding of PGZ to monitor the change of blood sugar.

  1. what was the body weight range (g) of animals that used in study, mention in study protocol?

Answer: Thanks for reviewer 1’s comment. The body weight range of animals is around 25-30 g, we have added this information in line 398, which are marked in yellow.

Reviewer 2 Report

The authors describe a beneficial effect of pioglitazone treatment on RGC degeneration in a mouse model of ischemic optic neuropathy. While the findings are potentially interesting, there are a number of concerns that would need to be addressed before this report would be ready for publication.

Major critiques:

  1. My main concern is that the authors do not have a control group in which non-diabetic mice (e.g. mice not treated with streptozotocin) are subjected to experimental ischemic optic neuropathy and receive pioglitazone or DMSO. We don’t know if the beneficial effect of pioglitazone that the authors report is due to a reduction of hyperglycemia or due to the pioglitazone itself (e.g. some downstream effect of PPAR-gamma). We know from the authors’ original paper (REF 13) that even non-diabetic mice suffer RGC loss after experimental ischemic optic neuropathy, so it should be possible to see if pioglitazone has a beneficial effect in these mice too. Without this experiment it is difficult to interpret the findings of this paper overall.
  2. Regarding the cell culture work (Figures 7 and 8), I don’t think that these experiments add anything to this paper. RGC-5 cells are quite controversial, in terms of whether their biology has any similarity to actual RGC cells. Also, bathing the cells in media with high glucose under hypoxia seems like a very artificial comparison to NAION, where it is the RGC axons that are deprived of blood flow, not the somas. Also, the RGC-5 viability assay in Fig 7 (worse in PGZ treated cells under 25 mM glucose than under 5.5 mM glucose) does not correlate with TUNEL staining (less severe in PGZ treated cells under 25 mM glucose than under 5.5 mM glucose). I think I would just scrap these experiments; the in vivo work is what really matters here.
  3. Regarding Fig 2. The original paper from this group (REF 13) describes thickening of the ganglion cell complex (GCC, which they define as RNFL + GCL + INL) soon after inducing ischemia, with GCC thinning apparent 3 weeks later. In that paper they also described a correlation between severity of initial swelling and the severity of the later atrophy. RNFL thickening is a hallmark of ischemic optic neuropathy, and since the GCC thickening was reported previously in their model, one would expect to see clear signs of it at Day 1 in the present manuscript. Comparing to humans with NAION, I am surprised that swelling would have resolved in these mice at week 1 (even week 3 would be early for humans, but more believable). Another issue is that it appears that the original paper performed a peripapillary (circular) OCT scan when measuring the GCC and retinal thickness, whereas the present work just did linear scans.  I think it is important that the authors reproduce the GCC thickening that occurs early after inducing ischemia, so that we can be sure the photochemical thrombosis worked.  The methods suggest that OCT was also performed at 3 weeks post-ischemia, so these thickness values should be reported rather than week 1. If a peripapillary OCT scan was performed, this would be a better source from which to derive thickness data.

 Other critiques:

--In the introduction, where the authors reference their prior work in this model (REF 13), they mistakenly call it a rat model, whereas the cited paper uses a mouse model, as in the present study.  In this sentence and the next, it would be more clear to call the strategy “glycemic control” than “hyperglycemia control.”

--In this manuscript, the Materials and Methods section is at the very end.  Thus, it is not appropriate to first define group 1, group 2, and group 3 there.  The Results section simply refers to these group numbers without any context, making it difficult to figure out what is going on. If the authors wish to use these group numbers throughout the Results section (rather than describing the treatments each time), they will need to define the groups at the very beginning of Results.

--Results 2.1: it states that the body weights increased but then gives a P value of 0.479, which does not approach statistical significance. Perhaps animal weight is not all that important to this paper.

--Fig 1B. What time point does this comparison represent, the final time point?  Not sure that 1B and 1D are necessary, since they seem to be taken directly from 1A and 1C.

--Fig 4. The magnification is too low to know if the green puncta truly represent mononuclear inflammatory cells.  We would need to see higher magnification images to see if the morphology is believable.

--Fig 5. Again, this is very low magnification. It is hard to tell exactly what the authors are pointing at.  It would seem that co-labeling for RGCs would be important, to build a case that PPAR-gamma is being up-regulated within RGCs themselves (or some other cell type, if the authors think that is relevant). Might also consider performing a Western blot on retina lysates to confirm an up-regulation of PPAR protein.  Also, the slightly higher magnification insets in the 2nd, 4th, and 6th rows are not explained in the legend.  Are they taken from the same image, and what magnification are they?

--Fig 6. The Brn3a staining is very faint and difficult to compare between groups.  The Brn3a staining in the authors’ previous paper (REF 13) is much higher quality. The authors should choose better representative images for this figure.

Round 2

Reviewer 2 Report

I appreciate the authors' improvements to Figures 5 and 6. However, I continue to think that the overall design of this study does not allow for a clear interpretation. It is entirely unclear whether it is the improvement of systemic glycemic control or some other PPAR-gamma-related signaling effect within RGCs that improves RGC survival in this in vivo AION model. Without a non-diabetic control, this cannot be known.  Furthermore, as I alluded to previously, AION is a disease in which the RGC axons are deprived of perfusion as they exit through the optic nerve head.  The RGC somas themselves are not infarcted (this would occur in a central retinal artery occlusion, an entirely different disease).  Therefore, it does not make sense to study RGC-5 cell bodies as a cell culture model of AION.  The fact that the authors point out that the cell culture results might reflect RGC-5 cell proliferation and cell cycle arrest further makes this point: RGCs in the adult retina do not proliferate/regenerate, so it is unclear how these cell culture findings would be relevant to AION.